# Antagonistic Interaction between Histone Deacetylase Inhibitor: Cambinol and Cisplatin—An Isobolographic Analysis in Breast Cancer In Vitro Models

**DOI:** 10.3390/ijms22168573

**Published:** 2021-08-09

**Authors:** Marta Hałasa, Jarogniew J. Łuszczki, Magdalena Dmoszyńska-Graniczka, Marzena Baran, Estera Okoń, Andrzej Stepulak, Anna Wawruszak

**Affiliations:** 1Department of Biochemistry and Molecular Biology, Medical University of Lublin, Chodzki 1 Street, 20-093 Lublin, Poland; marta.halasa@umlub.pl (M.H.); magdalena.dmoszynska-graniczka@umlub.pl (M.D.-G.); marzena.baran@umlub.pl (M.B.); estera.okon@umlub.pl (E.O.); andrzej.stepulak@umlub.pl (A.S.); 2Department of Pathophysiology, Medical University, Jaczewskiego 8 Street, 20-090 Lublin, Poland; jarogniew.luszczki@umlub.pl

**Keywords:** histone deacetylase inhibitors, cambinol, CDDP, combined therapy, isobolographic analysis

## Abstract

Breast cancer (BC) is the leading cause of death in women all over the world. Currently, combined chemotherapy with two or more agents is considered a promising anti-cancer tool to achieve better therapeutic response and to reduce therapy-related side effects. In our study, we demonstrated an antagonistic effect of cytostatic agent-cisplatin (CDDP) and histone deacetylase inhibitor: cambinol (CAM) for breast cancer cell lines with different phenotypes: estrogen receptor positive (MCF7, T47D) and triple negative (MDA-MB-231, MDA-MB-468). The type of pharmacological interaction was assessed by an isobolographic analysis. Our results showed that both agents used separately induced cell apoptosis; however, applying them in combination ameliorated antiproliferative effect for all BC cell lines indicating antagonistic interaction. Cell cycle analysis showed that CAM abolished cell cycle arrest in S phase, which was induced by CDDP. Additionally, CAM increased cell proliferation compared to CDDP used alone. Our data indicate that CAM and CDDP used in combination produce antagonistic interaction, which could inhibit anti-cancer treatment efficacy, showing importance of preclinical testing.

## 1. Introduction

Breast cancer (BC) is the most frequently diagnosed type of cancer in women worldwide, with a mortality rate of 15% [1]. BC is divided into molecular subtypes based on genetic and protein landspace, including HER2, estrogen (ER), progesterone (PR) receptors, and Ki-67 nuclear protein expression associated with cell proliferation [2,3]. Molecular profiling distinguishes five subtypes of breast cancer: luminal A (ER+/PR+, HER2−), luminal B (ER+/PR+, HER2− or HER2+), triple-negative BC, HER2-enriched (ER-/PR-HER2+), and normal-like (ER+/PR+, HER2−) breast cancer [3]. Triple negative BC (TNBC) is associated with aggressive clinical behavior resulting in poor patient prognosis. Due to the lack of ER and HER receptors expression, there are very limited options of targeted and immunotherapy for this type of cancer [4,5]; thus, standard chemotherapy (anthracyclines, alkylating agents, taxanes) still remains dominant in TNBC treatment [6]. Further serious limitations of chemotherapy in BC arise from acquired drug resistance, which is considered a key failure factor in anti-cancer treatment [7,8]. Chemoresistance is closely associated with genetic tumor burden and its rate of growth, immune response, and influence on tumor environment [9,10]. To overcome these obstacles, combined therapies are increasingly being used. A combination of two or more agents could be more efficient compared with drugs used separately. Moreover, this type of therapy is able to decrease the side effects in order to improve the patient’s life comfort. Combining chemotherapeutics with different mechanisms of action has a strong potential to improve patients’ outcomes [11]. An increasing number of studies combining epigenetic therapy with standard chemotherapy display promising results [12,13,14].

Histone deacetylase inhibitors (HDIs) are group of compounds that receive growing attention in the treatment of cancer in monotherapy, as well as in combination with other anti-cancer drugs [15,16]. Their targets, histone deacetylases (HDACs) belong to the class of enzymes acting on the epigenetic level, thus, influencing gene expression. HDACs are divided into four classes, including a third class named sirtuins. The sirtuins family consists of seven members, called SIRT1 to SIRT7, that deacetylate histones and non-histone proteins at specific sides and require nicotinamide adenine dinucleotide (NAD) for their activity [17]. Taking into account that SIRT1 is involved in BC development [18], HDIs seem to be promising agents in BC treatment [19].

Cambinol (CAM), a cell-permeable β-naphtol derivative, inhibits SIRT1 and SIRT2 activity (IC_50_ = 56 uM and IC_50_ = 59 uM, respectively) [20,21]. The mechanism of CAM action based on SIRT1 and SIRT2 inhibition leads to an increase of p53, FOXO3a, and Ku70 acetylation under stress condition; as a consequence, the cells become more sensitive to etoposide and paclitaxel, drugs commonly used in chemotherapy [22]. In addition to regulating FOXO3a-Ac level, CAM also affects FOXO1 acetylation and its expression in hepatocellular carcinoma (HCC) cells, which influences the ability of p53 to induce apoptosis [23]. Additionally, CAM alters HCC lines morphology; however, morphologic changes do not influence cell apoptosis nor caspase-3 activity [24]. To validate CAM activity against SIRT1, in vivo research was done using an HCC xenograft model [24,25]. The results indicated overall smaller size of tumor and decrease in vascular density after CAM treatment compared to vehicle treated control [25]. In experimental therapy, CAM enhances the cell response to sorafenib treatment, leading to a conclusion that HDI has an important role in ameliorating adverse effects through reduction of chemotherapeutic doses [26].

Platinum-based compounds are broadly applied for chemotherapy. Cisplatin (CDDP) is one of the derivatives of platinum used as a highly effective drug in a variety of cancers, such as sarcomas, lymphomas, and germ cell tumors and carcinomas, including BC [27]. The CDDP chemical structure consists of a doubly charged platinum ion surrounded by two amine ligands and two chloride ligands. Amine ligands strongly interact with platinum ions, while chloride ligands are exchanged during nucleophilic substitution with guanyl base nitrogen atoms in the DNA chain. Due to the ability to crosslink with DNA, CDDP initiates DNA lesions, activating the DNA damage response, and inducing the intrinsic apoptosis pathway [27,28]. The main limitation of CDDP treatment is resistance to platinum-based agents during cancer therapy. Moreover, CDDP exhibits serious adverse side effects, including nausea and vomiting, blood test abnormalities, and kidney toxicity [29]. Reducing these unwanted reactions, together with increasing the therapy’s efficiency is one of the main challenges of anti-cancer treatment. As we demonstrated previously [12,30,31], combined therapy with HDIs and CDDP resulted in significant reduction of both drugs’ dosages necessary to achieve a similar anti-cancer effect compared to single-drug treatment.

The aim of this study is to investigate the anti-cancer activity of CAM and CDDP administered alone or in combination on two types of BC cell lines (luminal BC: MCF7, T47D, and TNBC: MDA-MB-231, MDA-MB-468) using advanced pharmacokinetic isobolographic analysis.

## 2. Results

### 2.1. CAM and CDDP Used Separately Decrease Proliferation of MCF7, T47D, MDA-MB-231, and MDA-MB-468 BC Cell Lines

The anti-proliferative activity of CAM and CDDP for each BC cell line was assessed using an MTT assay. All BC cell lines were exposed to increasing concentrations of CAM or CDDP. CAM and CDDP administered separately induced inhibition of cell viability in each BC cell line in a concentration-dependent manner. The median inhibitory concentration (IC_50_ values ± S.E.M.) for each line was calculated based on long-probit analysis of the concentration–response relationship effects of both agents (Figure 1, Table 1).

### 2.2. Anti-Proliferative Effect of CAM and CDDP Administered Alone and Inhibition of This Effect after Treatment of CAM in Combination with CDDP in BC Cell Lines (MCF7, T47D, MDA-MB-231, and MDA-MB-468)

The antiproliferative effect of CAM and CDDP administered in combination against BC cell lines was assessed using MTT assay. CAM and CDDP were administered in combination with 1:1 drug mixture in increasing concentrations. All BC cell lines were exposed to the CAM and CDDP mixture treatment using different ratios of IC_50_ (2.0 means IC_50_ of CAM + IC_50_ of CDDP). The results are presented as median inhibitory concentrations (IC_50_ values in μg/mL ± S.E.M.) of CAM and CDDP administered in combination with respect to their anti-proliferative effects in four breast cancer cell lines (MCF7, T47D, MDA-MB-231, and MDA-MB-468) (Figure 2).

CAM and CDDP administered alone produced anti-proliferative effects in all BC cell lines. The equations of dose–response relationship curves (DRRCs) allowed us to determinate IC_50_ values for CDDP, which were 1.57 µg/mL (y = 1616x + 4.6829), 1.13 µg/mL (y = 1.0066x + 4.9479), 1.51 µg/mL (y = 0.9637x + 4.8285), and 1.57 µL/mL (y = 0.9564x + 4.8123) for MCF7, T47D, MDA-MB-231, and MDA-MB-468, respectively. Simultaneously, the equations of dose–response relationship curves (DRRCs) that allowed us to determine IC_50_ values for CAM were calculated: 20.31 µg/mL (y = 6.6232x − 3.6617), 16.51 µg/mL (y = 5.7965x − 2.0591), 13.44 µg/mL (y = 4.0016x + 0.485), and 15.03 µL/mL (y = 4.8046x − 0.6546) for MCF7, T47D, MDA-MB-231, and MDA-MB-468, respectively (Table 1).

The parallelism test of CRRCs between CAM and CDDP revealed that the DRRCs of both agents were non-parallel to each other (Table 1, Figure 3). For each cell line, the combination of CAM and CDDP at the fixed-ratio 1:1 inhibited anti-proliferative effects, and the IC_50_ values calculated from the DRRC for the mixture of CAM and CDDP were: 32.48 µg/mL (y = 7.3866x − 6.166), 19.11 µg/mL (y = 2.1449x + 2.2517), 21.35 µg/mL (y = 4.1335x − 0.4953), and 15.71 µg/mL (y = 3.5604x + 0.7408) for MCF7, T47D, MDA-MB-231, and MDA-MB-468, respectively (Figure 3).

### 2.3. Type I ISOBOLOGRAPHIC Analysis of Interaction for CAM and CDDP in MCF7, T47D, MDA-MB-231, and MDA-MB-468 Cell Lines

The combination of CAM and CDDP at a fixed-ratio of 1:1 induced inhibition of anti-proliferative effect in MCF7, T47D, MDA-MB-231, and MDA-MB-468 cell lines. The type of interaction between both agents was antagonism for MCF7 and MDA-MB-231 cells, and the obtained results were statistically significant. In T47D cell lines and MDA-MB-468 cells, treatment of CAM and CDDP in combination showed a tendency towards antagonism. The isobolographic interactions between CAM and CDDP for all cell lines are described in Table 2. The effects of combined treatment of CAM and CDDP in detail are presented in Figure 4.

### 2.4. CAM and CDDP Administered Separately Induce Cell Apoptosis, While the Administration in Combination Abolishes this Effect in MCF7, T47D, and MDA-MB-468 Cell Lines

The impact of CAM and CDDP administered separately and in combination on induction of apoptosis was measured as a number of cells with activated caspase-3 (Figure 5). CAM and CDDP were used in a concentration of IC_50_ or 2IC_50_. The data obtained from isobolographic analysis were used to select the appropriate concentration of both agents. CAM and CDDP used separately induced an increase of the cells number with active caspase-3. CAM and CDDP used in combination significantly decreased the percentage of cells with active caspase-3, suggesting that CAM abolishes the effect of CDDP. Therefore, the antagonistic interaction calculated by isobolographic analysis was confirmed by assessment of apoptosis for luminal A subtype (MCF7 and T47D) and MDA-MB-468 cell lines. In T47D and MCF7 cells, after 2IC_50_ CAM treatment, the active caspase-3 had not been decreased meaningfully compared to IC_50_. In MCF7 cells, the 2IC_50_ decrease was not statistically significant compared to the control. This phenomenon may be explained by the fact that CAM does not activate caspase-3, whereas CDDP does, and CAM abolishes the CDDP effect on caspase-3, which leads to a decrease in active caspase-3.

Interestingly, in the MDA-MB-231 cell line, the apoptosis induction caused by CAM and CDDP used separately and both agents used in combination remained at the same level for IC_50_ concentration. For 2IC_50_ concentration, there was a slight increase in the number of cells with active caspase-3 for CAM and CDDP used in combination, compared with drugs used separately. The difference in the level of active caspase-3 between CAM and CDDP used alone and combined is weak, which may confirm that only CDDP induces caspase-3 activation.

### 2.5. The Impact of CAM and CDDP used Separately and in Combination on Cell Cycle Progression for MCF7, T47D, MDA-MB-231, and MDA-MB-468 Cell Lines

The impact of CAM and CDDP administered separately and in combination on the cell cycle is presented in Figure 6. CAM and CDDP were used separately and in combination to assess the cycle progression. For each analyzed cell line, CDDP administered alone in IC_50_ and 2IC_50_ concentration led to cell arrest in phase S; however, the number of cells arrested in S-phase was slightly higher for TNBC lines. CAM abolished this effect in each BC cell line.

According to the cell cycle progression analysis, the cell cycle changes induced by CAM and CDDP administered separately and in combination were not dependent on the type of cell line.

## 3. Discussion

Sirtuins are known as key players in tumor development affecting a variety of signaling pathways [34,35]. Among all members of the sirtuin family, SIRT1 and SIRT2 have been extensively studied due to their prominent role in cancer cell fate [36,37]. They are regarded as tumor suppressors or oncoproteins depending on the type of cancer [38], which emphasizes the importance of the search for new sirtuins modulators. So far, no sirtuin inhibitor has been approved by the Food and Drug Administration (FDA) for treatment [39]; however, some of those compounds seem to be promising tools in anti-cancer therapy in view of preclinical studies [40,41]. Several reports have confirmed the utility of sirtuin inhibitors in combined anti-cancer therapy in vitro and in vivo (26,42,43). For example, downregulation of SIRT1 expression by EX527 enhanced cell sensitivity to bortezomib in multiple melanoma [42], while SIRT1 inhibition by tenovin-6 in combination with tyrosine kinase inhibitor imatinib targeted leukemia stem cells, leading to reduction of their growth [43].

In our study, we analyzed the interaction between CAM, SIRT1 and SIRT2 inhibitor, and the CDDP standard chemotherapeutic used in BC therapy. We analyzed four BC cell lines differing from each other by molecular profile [44]. Our findings demonstrated that CAM inhibits cell proliferation in a dose-dependent manner in each analyzed BC cell line. These results remain in line with previously-reported findings for HCC [24], neuroblastoma [45], and multiple melanoma cell lines [46]. Our results indicate that the MCF7 luminal A cell line was the most resistant to CAM treatment, while TNBC cell lines were the most sensitive to CAM treatment. An identical observation was made when the cells were incubated with CDDP. The MCF7 cell line was the most resistant, while the other analyzed cell lines showed higher sensitivity for CDDP treatment.

In multiple melanoma cells, CAM inhibits cell proliferation in a time- and dose-dependent manner, increasing apoptosis and arresting cells in G1 phase [46]. In our study, CAM did not significantly affect cell cycle progression; however, it slightly increased caspase-3 activation in a dose-dependent manner compared with control. In other experiments, CDDP reduced cell viability and induced apoptosis in each BC cell line in a dose-dependent manner, which is in agreement with other reports indicating that [47] CDDP could affect cell apoptosis through caspase-3 activation.

Although the molecular mechanism of CDDP action and its cytotoxic implication is well understood, there is still need to search for appropriate drugs that will allow us to reduce its cytotoxicity while maintaining therapeutic benefits [48]. Our findings showed that CAM and CDDP used separately inhibited cell proliferation for each BC cell line; however, the combination of those agents abolished this effect, indicating antagonistic interaction. The possible reason for the observed phenomenon could be explained by an overlapping mechanism of action of CDDP and CAM. Since CAM influences p53, FOXO3a reversible acetylation, and consequently, stabilization of these apoptosis-related proteins, whereas with CDDP, p53 is a pivotal determinant of cisplatin sensitivity [49], and upregulated FoxO3a expression enhances CDDP sensitivity [50], these activities could abolish each other. Antagonistic interaction between CAM and CDDP could be also considered based on another finding [51], indicating that CDDP induced SIRT1 expression in HEK 293 cell lines. SIRT1 overexpression protects against CDDP-induced cellular damage [51]. Due to this observation and given the fact that CAM inhibits SIRT1 activity, the mechanism of antagonistic interaction between those two drugs could be a result of simultaneous, opposite action against SIRT1 activity. On the other hand, it has been demonstrated [52] that SIRT1 is involved in the acquisition of CDDP-resistance in oral squamous cell carcinoma (OSCC) cells. SIRT1 overexpression significantly discourages cell growth inhibition and induces apoptosis. The use of a sirtuin inhibitor dramatically abolishes SIRT1-mediated CDDP resistance in OSCC cells, indicating that CDDP resistance partly arises from SIRT1 deacetylase activity [52].

It is obvious that synergetic interaction is the most favorable; however, drugs often interact in a non-synergistic way [53,54], which should be also reported. In recent years, the assumption that synergy between drugs translates directly into clinical benefit has been undermined [53,54]. Synergic interaction between two drugs does not ensure that inhibition of proliferation will occur for those cells whose proliferation has not previously been stopped by drugs used separately. Correspondingly, an antagonistic interaction is not always clinically adverse and unfavorable for some drug combinations [55]. Our results are commensurate with previous research [56], where CAM in combination with doxorubicin (DOX) did not show a significant synergetic effect for neuroblastoma cells. Although CAM administered alone is effective in DOX-sensitive cells, no major synergetic effect was observed after administration of CAM in combination with DOX [56]. In contrast, it has to be emphasized that CAM is able to give a synergetic effect in HCC cancer cells in combination with sorafenib [26], which is a targeted drug approved for patients with differentiated thyroid cancer [57]. These agents in combination induce cell apoptosis through activating proapoptotic proteins compared to treatment by sorafenib alone [26]. This synergetic interaction could occur due to selecting different molecular targets. All these observations bring us to the conclusion that CAM has chemotherapeutic potential and should be tested with chemotherapeutics with different mechanisms of action than CDDP. Anthracyclines and taxanes used in BC therapy present different mechanisms [58,59] and, thus, could be good candidates for preclinical testing.

Taken together, our data indicate that CAM and CDDP used in combination produce antagonistic interaction. The type of pharmacological drug–drug interaction between CAM and CDDP disclaims potential application of combined treatment in BC cells; however, this interaction could be completely different in other cancer cell lines in which SIRT1 and SIRT2 are involved and in combination with different anti-cancer drugs.

## 4. Materials and Methods

### 4.1. Drugs

Cambinol (CAM) was purchased from Sigma-Aldrich (St. Louis, MO, USA). A stock solution of CAM (100 mM) was prepared in dimethyl sulfoxide (DMSO). Cisplatin (CDDP) was purchased from Sigma (St. Louis, MO, USA) and dissolved in phosphate buffered saline (PBS) without Mg^2+^ and Ca^2+^ at 1 mg/mL as a stock solution. The drugs were diluted with culture medium to the respective concentration just before use.

### 4.2. Cell Lines

MCF7 (ATCC^®^ HTB-22^TM^), T47D (ATCC^®^ HTB-133^TM^), MDA-MB-231 (ATCC^®^ CRM-HTB-26^TM^), and MDA-MB-468 (ATCC^®^ HTB-132^TM^) human BC cell lines were obtained from the American Type Culture Collection (Manassas, VA, USA). T47D, MCF7, MDA-MB-231, and MDA-MB-468 cancer cells were grown in DMEM/Nutrient F-12 Ham (DMEM/F12) culture medium (Sigma) with 10% FBS (Sigma), penicillin (100 μg/mL), (Sigma) and streptomycin (100 μg/mL) (Sigma). Mycoplasma-free cultures were maintained in a humidified atmosphere with 5% CO_2_ at 37 °C.

### 4.3. Cell Viability Assay

MCF7, T47D, MDA-MB-231, and MDA-MB-468 BC cell lines were placed on a 96-well plate (Nunc, Rochester, New York, NY, USA) at a density of 1 × 10^4^ cells/mL. The next day, the culture medium was removed, and cells were exposed to serial dilutions of total concentration of CAM (10–100 µM) or CDDP (0.01–10 µg/mL) individually or a combination of both compounds for 96 h. Then, the BC cell lines were incubated with the 3-(4,5-dimethylthiazol-2-yl)-2,5-diphenyltetrazolium bromide (MTT) solution at a concentration of 5 mg/mL (Sigma) for 3 h. During this time, MTT was metabolized by living cells to purple formazan crystals, which were solubilized in a sodium dodecyl sulfate (SDS) buffer (10% SDS in 0.01 N HCl) overnight. The optical density of the product was measured at 570 nm using an Infinite M200 Pro microplate reader (Tecan, Männedorf, Switzerland). Dose–response curves were plotted to determine half-maximal inhibitory concentrations (IC_50_) for the CAM and CDDP using GraphPad Prism 9.0 (GraphPad Software, San Diego, CA, USA). The results of combined treatment CAM and CDDP were analyzed according to isobolographic protocol. The drug doses were determined based on the IC_50_ values calculated from the previous cytotoxicity test.

### 4.4. Assessment of Apoptosis

MCF7, T47D, MDA-MB-231, and MDA-MB-468 cell lines were treated with tested compounds individually or in combination for 48 h, and as a control, cells without treatment were used. Afterwards, cells were harvested, fixed, and permeabilized using the Cytofix/Cytoperm. All experiments were performed according to the manufacturer’s instructions of PE Active Caspase-3 Apoptosis Kit (BD Biosciences, San Jose, CA, USA). Labeled cells were analyzed by flow cytometer FACSCalibur (Becton Dickinson, Franklin Lakes, NJ, USA), operating with CellQuest software (BD Biosciences) to quantitatively assess the caspase-3 activity.

### 4.5. Cell Cycle Analysis

MCF7, T47D, MDA-MB-231, and MDA-MB-468 cell lines were treated with tested compounds and mixture for 48 h and then fixed in 80% ethanol at −20 °C for 24 h. The experiment was conducted by utilizing PI/RNase Staining Buffer (BD Biosciences) according to the manufacturer’s instructions. Cell cycle analysis was performed using flow cytometry FACSCalibur (Becton Dickinson). Acquisition rate was 60 events per second in low acquisition mode, and 10,000 events were measured.

### 4.6. Isobolographic Analysis of Interactions for Parallel and Nonparallel Concentration–Response Relationship Effects of CAM and CDDP

The pharmacodynamic nature of interactions between CAM and CDDP in four BC cell lines was analyzed by means of the type I isobolographic analysis for both parallel and nonparallel concentration–response relationship lines [60,61]. To start the isobolographic analysis of interaction between CAM and CDDP, the inhibition of cell viability of all BC cell lines was determined in the MTT test, as described in 4.3. Due to the log-probit method, it was possible to determine median inhibitory concentrations (IC_50_ values) for CAM and CDDP in each cell line, as recommended elsewhere [30,31]. The median additive inhibitory concentrations (IC_50 add_) for drugs in combination was theoretically calculated using the method described elsewhere [30]. The calculated values were used for performing MTT for all BC cell lines—the assessment of experimentally derived IC_50 mix_ values for tested drug combinations in a fixed 1:1 ratio. The particular drug concentrations (CAM and CDDP) in combination were calculated by multiplying IC_50 mix_ values accordingly to proportions in additive drug combinations by means of the unpaired Student’s *t*-test with Welch’s correction, as recommended elsewhere [62,63]. An isobolographic method was described in detail elsewhere [30,31].

### 4.7. Statistical Analysis

The IC_50_ and IC_50_ mix values for CAM and CDDP administered separately or in combination at the fixed ratio of 1:1 were calculated by computer-assisted log-probit analysis according to Litchfield and Wilcoxon [64]. The experimentally-derived IC_50_ exp values for the combination of CAM and CDDP were statistically compared to their respective theoretical additive IC_50_ add values by the use of unpaired Student’s *t*-test with Welch’s correction, according to Tallarida [63]. Results were analyzed using GraphPad Prism 9.0 software (one-way ANOVA; Tukey’s post-hoc testing). Statistical differences were considered relevant at *p* < 0.05 (* *p* < 0.05, ** *p* < 0.01, *** *p* < 0.001). Data are expressed as mean ± standard deviation of the mean (±SD).

## Figures and Tables

**Figure 1 ijms-22-08573-f001:**
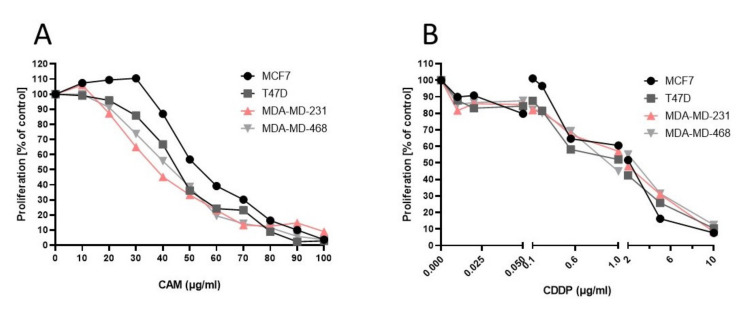
The antiproliferative effect of CAM and CDDP used separately against BC cell lines (**A**,**B**). Inhibition of cell proliferation was measured by the MTT assay after 96 h of treatment with various concentrations of active agent. All results were analyzed using GraphPad Prism 9.0 software (ordinary one-way ANOVA; Dunnett’s multiple comparisons test). Statistical differences were considered relevant at *p* < 0.05 (* *p* < 0.05, ** *p* < 0.01, *** *p* < 0.001 versus the control). Data are expressed as mean ± standard deviation of the mean (±SD).

**Figure 2 ijms-22-08573-f002:**
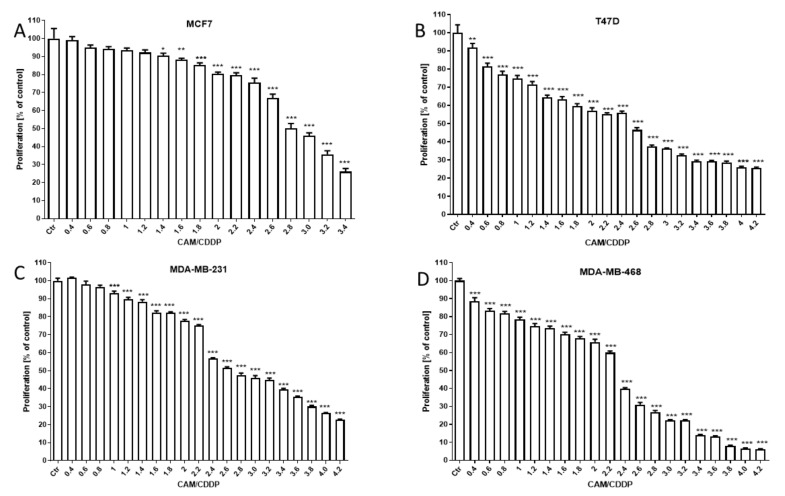
The antiproliferative effect of CAM and CDDP administered in combination against BC cell lines (**A**–**D**). Inhibition of cell proliferation was measured by the MTT assay after 96 h treatment with various concentrations of active agents. The antiproliferative effect of CAM and CDDP administered in combination with 1:1 drug mixture in increasing concentrations. All BC cell lines were exposed to CAM and CDDP mixture treatment using different ratios of IC_50_ (2.0 means IC_50_ of CAM + IC_50_ of CDDP). All results were analyzed using GraphPad Prism 9.0 software (ordinary one-way ANOVA; Dunnett’s multiple comparisons test). Statistical differences were considered relevant at *p* < 0.05 (* *p* < 0.05, ** *p* < 0.01, *** *p* < 0.001 versus the control). Data are expressed as mean ± standard deviation of the mean (±SD).

**Figure 3 ijms-22-08573-f003:**
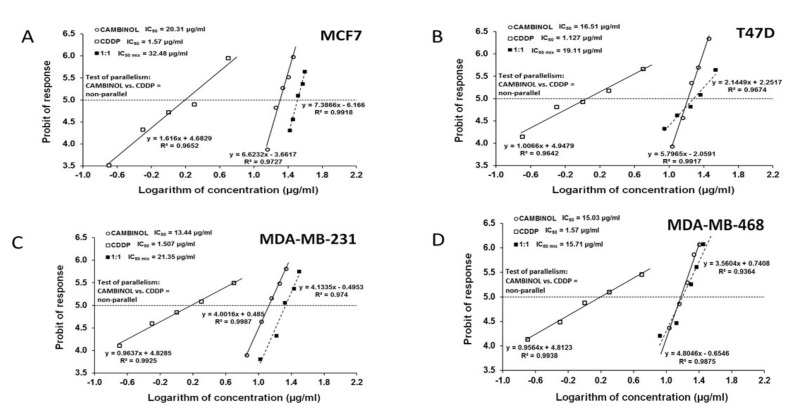
The log-probit dose–response relationship lines for CAM and CDDP in MCF7, T47D, MDA-MB-231, and MDA-MB-468 cell lines. (**A**) Log-probit dose–response relationship lines for CAM and CDDP administered alone and in combination at the fixed-ratio of 1:1, with respect to their anti-proliferative effects on the MCF7 cell lines. (**B**) Log-probit dose–response relationship lines for CAM and CDDP administered alone and in combination at the fixed-ratio of 1:1, with respect to their anti-proliferative effects on the T47D cell lines. (**C**) Log-probit dose–response relationship lines for CAM and CDDP administered alone and in combination at the fixed-ratio of 1:1, with respect to their anti-proliferative effects on the MDA-MB-231 cell lines. (**D**) Log-probit dose–response relationship lines for CAM and CDDP administered alone and in combination at the fixed-ratio of 1:1, with respect to their anti-proliferative effects on the MDA-MB-468 cell lines.

**Figure 4 ijms-22-08573-f004:**
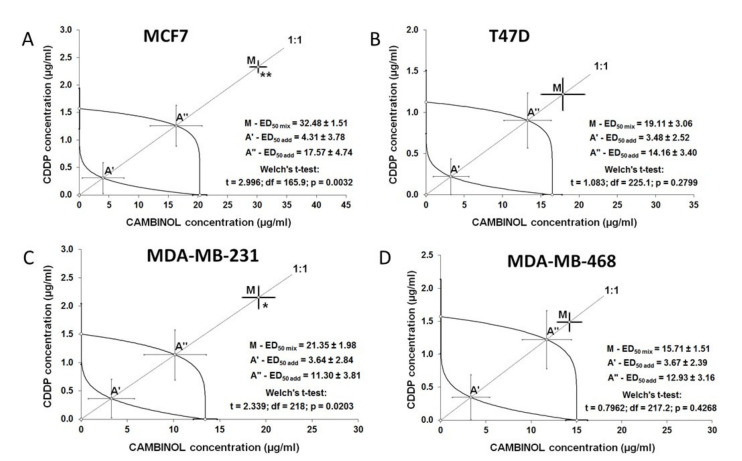
Isobolograms illustrating types of interactions between CAM and CDDP in all BC cells. The IC_50_ ± S.E.M. for CAM and CDDP are plotted graphically on the X- and Y-axes, respectively. The lower and upper isoboles of additivity represent the curves connecting the IC_50_ values for CAM and CDDP administered alone. The points A′ and A″ depict the theoretically calculated IC_50_ add values (±S.E.M.) for both lower and upper isoboles of additivity. The point M on each graph represents the experimentally-derived IC_50_ mix value (±S.E.M.) for the total dose of the mixture, which produced a 50% anti-proliferative effect in the BC cell lines. (**A**) The antagonism interaction between CAM and CDDP with respect to their anti-proliferative effects in MCF7 cell lines measured in vitro by the MTT assay. The experimentally-derived IC_50 mix_ value is placed outside the area of additivity and indicates antagonism interaction. (**B**) CAM and CDDP in combination showed tendency towards antagonism with respect to their anti-proliferative effects in T47D cell lines measured in vitro by the MTT assay. The experimentally-derived IC_50_ mix value is placed outside the area of additivity, closer to point A″ for the upper isobole of additivity than in MCF7 cells, which indicates tendency towards antagonism. (**C**) The antagonism interaction between CAM and CDDP with respect to their anti-proliferative effects in MDA-MB-231 cell lines measured in vitro by the MTT assay. The experimentally-derived IC_50 mix_ value is placed outside the area of additivity and indicates antagonism interaction. (**D**) The tendency towards antagonism between CAM and CDDP with respect to their anti-proliferative effects in MDA-MB-468 cell lines measured in vitro by the MTT assay. The experimentally-derived IC_50_ mix value is placed outside the area of additivity, close to the point A″ for the upper isobole of additivity, which indicates tendency towards antagonism. * *p* < 0.05, ** *p* < 0.01.

**Figure 5 ijms-22-08573-f005:**
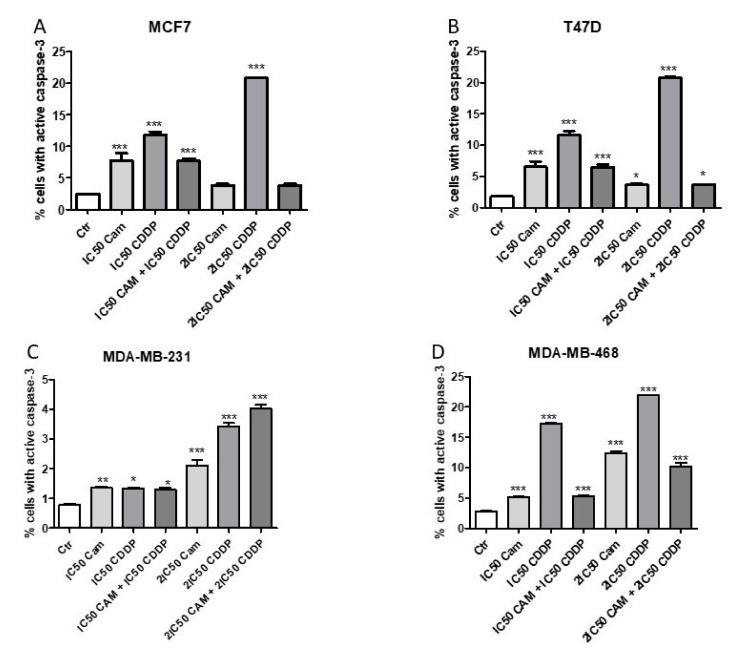
The effect of CAM and CDDP administered separately and in combination used in concentration of IC_50_ and 2IC_50_ for (**A**) MCF7, (**B**) T47D, (**C**) MDA-MB-231, and (**D**) MDA-MB-468 on cell lines on caspase-3 activation. All results were analyzed using GraphPad Prism 9.0 software (ordinary one-way ANOVA; Dunnett’s multiple comparisons test). Statistical differences were considered relevant at *p* < 0.05 (* *p* < 0.05, ** *p* < 0.01, *** *p* < 0.001 versus the control). Data are expressed as mean ± standard deviation of the mean (±SD).

**Figure 6 ijms-22-08573-f006:**
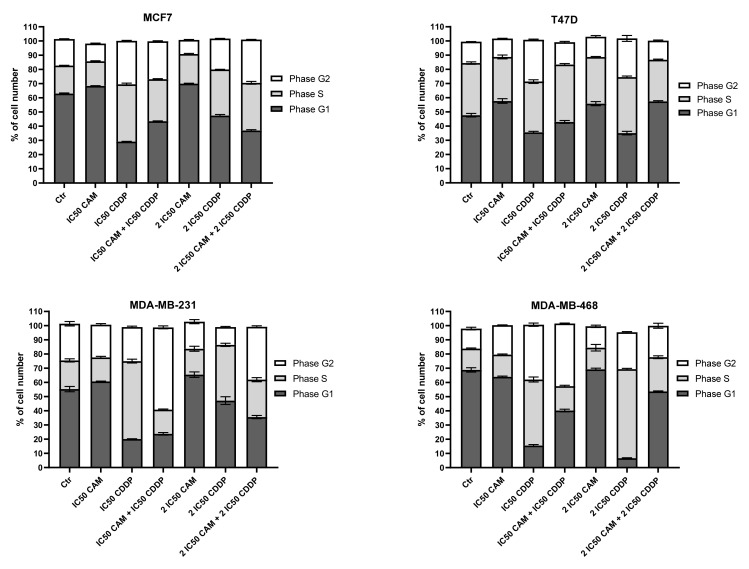
The effect of CAM and CDDP administered separately and in combination used in concentration of IC_50_ and 2IC_50_ for (**A**) MCF7, (**B**) T47D, (**C**) MDA-M23, and (**D**) MDA-MB-468 cell lines on cell cycle progression. The results are presented as mean ± SD from three independent experiments.

**Table 1 ijms-22-08573-t001:** Anti-proliferative effects of CDDP and CAM administered separately in BC cell lines (MCF7, T47D, MDA-MB-231, and MDA-MB-468) measured in vitro by the MTT assay. *n—*total number of items used at concentrations whose expected anti-proliferative effects ranged between 4 and 6 probits (16 and 84%).

Cell Line	Drug	IC_50_ (μg/mL)	*n*
MCF7	CDDP	1.571 ± 0.373	72
MCF7	CAMBINOL	20.314 ± 1.176	72
T47D	CDDP	1.127 ± 0.384	90
T47D	CAMBINOL	16.512 ± 1.261	54
MDA-MB-231	CDDP	1.507 ± 0.536	90
MDA-MB-231	CAMBINOL	13.437 ± 1.287	72
MDA-MB-468	CDDP	1.571 ± 0.563	90
MDA-MB-468	CAMBINOL	15.028 ± 1.199	72

**Table 2 ijms-22-08573-t002:** Isobolographic interactions between CDDP and CAM at the fixed-ratio combination of 1:1 in four cancer cell lines (MCF7, T47D, MDA-MB-231, and MDA-MB-468) measured in vitro by the MTT assay. Results are presented as median inhibitory concentrations (IC_50_ values in μg/mL ± S.E.M.) for two-drug mixtures, determined either experimentally (IC_50 mix_) or theoretically calculated (IC_50 add_) from the equations of additivity [32,33], blocking proliferation in 50% of tested cells in four cancer cell lines (MCF7, T47D, MDA-MB-231, and MDA-MB-468) measured in vitro by the MTT assay.

Cell Line	IC_50 mix_ (μg/mL)	*n* _mix_	^L^ IC_50 add_ (μg/mL)	*n* _add_	^U^ IC_50 add_ (μg/mL)	*n* _add_	*p*-Value	Type of Interaction
MCF7	32.483 ± 1.508	90	4.314 ± 3.781	140	17.573 ± 4.743	140	0.0032 **	antagonism
T47D	19.112 ± 3.055	90	3.478 ± 2.515	140	14.158 ± 3.404	140	0.2799	tendency towards antagonism
MDA-MB-231	21.352 ± 1.980	72	3.638 ± 2.840	158	11.304 ± 3.813	158	0.0203 *	antagonism
MDA-MB-468	15.713 ± 1.513	90	3.671 ± 2.391	158	12.927 ± 3.155	158	0.4268	tendency towards antagonism

*n*_mix_—total number of items used at those concentrations whose expected anti-proliferative effects ranged between 16% and 84% (i.e., 4 and 6 probits) for the experimental mixture; *n* _add_—total number of items calculated for the additive mixture of the drugs examined; ^L^—IC_50 add_ value calculated from the equation for the lower line of additivity; ^U^—IC_50 add_ value calculated from the equation for the upper line of additivity. Statistical evaluation of data was performed with unpaired *t*-test with Welch’s correction. * *p* < 0.05 and ** *p* < 0.01 vs. the respective IC_50 mix_ value.

## Data Availability

Data available in a publicly accessible repository.

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
