# Peer review of "Antagonistic Interaction between Histone Deacetylase Inhibitor: Cambinol and Cisplatin—An Isobolographic Analysis in Breast Cancer In Vitro Models"

_ijms, 2021, doi:10.3390/ijms22168573_

Round 1
Reviewer 1 Report
The authors presented the antagonistic effects of cambinol and cisplatin in breast cancer cell lines. Although research on therapeutic agents that can produce synergistic effects while reducing side effects in combination therapy research is active, DDI research between two therapeutic agents is also important, so this manuscript is very interesting. However, some revisions are required.
- The authors do not explain why they chose cambionl to test the antagonistic effect with CDDT. Because CDDT is a common breast cancer treatment, it can be studied in combination with other treatments. In the case of CAM, it is necessary to explain why only CAM was selected from among various HDAC inhibitors to achieve a combined effect.
- Also, CAM is one of class III HDACi. The results will be more meaningful if it is sorted out whether both CDDT and various other HDACi exhibit antagonism in the combined efficacy or only class III (SIRT1-related) antagonistic activity.
- Please edit lines 207 and 208. The dramatic reduction in apoptosis when CAM and CDDT was used in combination was compared to that of CDDT alone. The current sentence is not clear in its meaning.
- I think that additional data are needed to increase the novelty of the manuscript. Although antagonism is significant in elucidating DDI, the mechanisms why CAM inhibits the effects of CPPT have not been clearly identified. Therefore, in-depth studies of these mechanisms should be added.
Reviewer 2 Report
The manuscript titled “Antagonistic interaction between histone deacetylase inhibitor: cambinol and cisplatin – an isobolographic analysis in breast cancer in vitro models.” explores the combined effects of cambinol and cisplatin on various breast cancer cell lines. The authors have used various in vitro biochemical assays to test their hypothesis. The authors have demonstrated their hypothesis through carefully designed experiments and results. Following are questions that the authors should consider addressing.
- Consider presenting the MTT assay results of cambinol in Fig 1A in μg/mL instead of μM as the IC50 results were presented in the same units.
- We acknowledge the thought of the authors to present negative results. However, cambinol is an investigational drug that is neither tested in any of the clinical trials nor approved for the treatment of any disease indication. What is the rationale to select cambinol in the study? Further, there are different histone deacetylase inhibitors (Vorinostat, Romidepsin, Panobinostat, and Belinostat) that are approved for the treatment of various cancers. The results would be clinically relevant if any of the above-mentioned compounds were used in the study instead of cambinol.
- Present a brief explanation as to why cells (MCF7 and T47D) with active caspase-3 were decreased when double the IC50 concentration of cambinol was used. Also, provide an explanation for the increase in the number of cells with active caspase-3 when CAM and CDDP were used in combination compared to the individual treatment.
Round 2
Reviewer 1 Report
The authors responded appropriately to the comments.